

# Future changes in atmospheric rivers over East Asia under stratospheric aerosol intervention

Ju Liang[1], Jim Haywood[1,2]

[1] College of Engineering, Mathematics and Physical Sciences, University of Exeter, Exeter, EX4 4QE, UK
[2] Met Office Hadley Centre, Exeter, EX1 3PB, UK

*Correspondence to*: Ju Liang (J.Liang@exeter.ac.uk)

**Abstract.** Atmospheric rivers (ARs) are closely associated with historical extreme precipitation events over East Asia. The projected increase in such weather systems under global warming has been extensively discussed in previous studies, while the role of stratospheric aerosol, particularly for the implementation of stratospheric aerosol intervention (SAI), in such a
change remains unknown. Based on an ensemble of the UK Earth System Model (UKESM1) simulations, here we investigate changes in the frequency of ARs and their associated mean and extreme precipitation under a range of climate forcing, including greenhouse gas emission scenarios of high (SSP5-8.5) and medium (SSP2-4.5) levels, the deployment of SAI geoengineering (G6sulfur) and solar dimming (G6solar). The result indicates a significant increase in AR frequency and AR-related precipitation over most of East Asia in a warmer climate and the most pronounced changes are observed in
southern China. Comparing to G6solar and both the SSPs scenarios, the G6sulfur simulations indicate that SAI is effective in partly ameliorating the increases in AR activity over the subtropical region; however, it may result in more pronounced increases in ARs and associated precipitation over the mid-high latitude regions, particularly northeastern China and Japan. Such a response is associated with the further weakening of the mid-latitude westerly jet stream under SAI that favours the high-latitude AR activity. This is driven by the decreased meridional gradient of thermal expansion in the mid-high
troposphere associated with aerosol cooling across the tropical region, though SAI effectively ameliorates the widespread increase in thermal expansion under climate warming. Such a side effect of SAI over the populated region implies that caution must be taken when considering geoengineering approaches to mitigating hydrological risk under climate change.

## 1 Introduction

East Asia is a populated region exposed to devastating and frequent hydrological extremes due to the influence of
precipitating weather systems. One of such systems is that of atmospheric rivers (ARs) that feature elongated and intense moisture transport in the low troposphere. Numerous studies have suggested close linkages between ARs and extreme precipitation events over East Asia (Kamae et al., 2017a; 2017b; Pan and Lu, 2019; Liang et al., 2020; Fu et al., 2021; Kim et al., 2021; Park et al., 2021; Liang and Yong, 2021; 2022; Liang et al., 2022; Mo et al., 2022). The AR-precipitation association stems from different dynamical factors. First, the main AR pathway connects the mid-high latitudes of East Asia





with the subtropical moisture source regions, including the Arabian Sea, Bay of Bengal and South China Sea (Pan and Lu, 2019). Second, despite the abundant moisture within the AR plumes, the adjacent environment of ARs, in terms of precipitation efficiency, exhibits a relatively high efficiency for depleting atmospheric moisture to generate precipitation (Liang and Yong, 2021). This is dependent on the active synoptic-scale transient eddies in the downstream area of ARs, which further transport moisture to the mid-high latitudes and trigger precipitation through various physical processes such

as the release of low-level potential instability and orographic lifting (Gimeno et al., 2016; Kamae et al., 2021; Park et al., 2021). Given the AR-precipitation linkage, the response of ARs to future climate warming might have implications for the regional climate in East Asia and this has attracted studies on projecting the possible future changes in ARs over this region. The first attempt to investigate the possible future changes in ARs over East Asia was reported by Espinoza et al. (2018) using an ensemble of coupled GCMs, which projected a pronounced increase in the frequency of ARs affecting southern

China under the high-level greenhouse gas (GHG) emission scenario of the Representative Concentration Pathway (RCP8.5). Under the same scenario, more recent studies based on a high-resolution GCM projected similar results (Kamae et al., 2021). These similarities suggest that there is growing recognition of the projected increase in AR activity over East Asia under climate warming and such a change is not simply a function of the choice of methodology, although some studies, especially those based on the Tier-2 experiments of the Atmospheric River Tracking Method Intercomparison Project

(ARTMIP, Payne et al., 2020; O'Brien et al. 2022), have suggested that the projected changes in ARs are susceptible to uncertainties in the algorithm used for detecting ARs and the choice of climate models.

ARs can cause significant economic losses by triggering hydrological disasters such as flooding (Dominguez et al., 2018; Corringham et al., 2019), landslides (Cordeira et al. 2019; Miller et al., 2018) and frozen precipitation (Guan et al., 2016;

Liang and Sushama, 2019). In addition, ARs are associated with the melting of glaciers (Neff, 2018) and the weakening of ice shelf stability (Wille et al., 2022) due to the intense meridional transport of both sensible and latent heat within AR plumes (Shields et al., 2019; Liang and Yong, 2022b). Hence, the projected increase in ARs by the current climate models, particularly over the populated regions in East Asia, implies more catastrophic hydrological extremes and irreversible changes to high-latitude ecosystems in future climate that are badly in need of mitigation and adaptation strategies. Despite

the demands of enhancing early warning systems and climate adaptation planning to cope with the possible future changes in ARs, a better understanding of the effects of warming mitigation strategies including reduction of GHG emissions and geoengineering on AR climatology is required at present.

In the current context of technical capacity constraints and geopolitical factors causing increasing difficulties to achieve the

target of the Paris 21$^{st}$ Conference of Parties, i.e. the global mean temperature targets of 1.5 and 2°C above pre-industrial (e.g. Millar et al., 2017; IPCC, 2018), dramatic increases in the frequency of very extreme precipitation have been observed in the recent period (Myhre et al., 2019). This calls for research into unconventional mitigation strategies (e.g. MacMartin et al., 2018) to mitigate risks of future extreme precipitation changes. One of the most prominent approaches is the solar





radiation management strategies via stratospheric aerosol intervention (SAI), i.e. injection of the precursor of scattering aerosols (sulfur dioxide) into the stratosphere where the atmosphere is relatively stable and has a longer lifetime compared to that in the troposphere so as to achieve cooling of the planet (Lawrence et al., 2018). Among the concerns regarding SAI deployment are impacts of on Antarctic ozone recovery (e.g., Tilmes et al., 2008; Heckendorn et al., 2009), the termination effect (e.g. Jones et al., 2013), continued ocean acidification (e.g. Williamson and Turley, 2012), impacts on key modes of climate variability such as the North Atlantic Oscillation and Quasi-Biennial Oscillation (Jones et al., 2022), and moral and ethical issues surrounding any deployment (e.g. Lawrence et al., 2018). However, climate model simulations indicated that continuous SAI can effectively maintain the global surface temperature under the 2020 conditions (Tilmes et al., 2021), and lead to much ameliorated hydrological extremes than in unmitigated climate change scenarios (e.g. Jones et al., 2018). To provide physical assessments of the impact of SAI, the Geoengineering Model Intercomparison Project (GeoMIP, Kravitz et al., 2011; 2016; Xia et al., 2017) has performed a set of model experiments based on global climate models (GCMs) to simulate climate scenarios with injections of sulfur into the stratosphere. In East Asia, the G4 experiments (i.e. future climate simulations under the RCP4.5 scenario with constantly injecting sulfur dioxide into the low-level equatorial stratosphere at a rate of 5 Tg per year, Kravitz et al., 2011) of GeoMIP have been used to assess the role of SAI in ameliorating the changes in precipitation (Liu et al.,2021). However, the G4 experiments are based on an idealized one-point injection of sulfur dioxide at the equator with an abrupt beginning and termination that are the only represented potential deployment of geoengineering (Visioni et al., 2021).

As GeoMIP experiments progressed from G4 to G6, a new experiment (G6sulfur, Kravitz et al., 2015) considered the more recently developed GHG emission scenarios from the Shared Socioeconomic Pathway (SSPs, O'Neill et al., 2016) and more plausible SAI deployment, i.e. the injection of sulfur dioxide between 10°N and 10°S and between 18 and 20 km altitude. Similar SAI simulations have been proposed by the Stratospheric Aerosol Geoengineering Large Ensemble (GLENS, Tilmes et al., 2018) project. GeoMIP G6 also performed experiments of future climate simulations considering the reduction of the total incoming solar irradiance (G6solar, Visioni et al., 2021) and examining differences between G6sulfur and G6solar helps understand the role of aerosol-climate interaction, and in particular the impacts that aerosol-induced stratospheric heating may have on atmospheric dynamics under SAI and the uncertainties in its simulation using GCMs. Among others, GeoMIP experiments have investigated changes in precipitating weather systems including tropical cyclones (Jones et al., 2017; Wang et al., 2018) and extratropical cyclones (Gertler et al., 2020). For ARs, the recent study based on the GLENS experiments found that SAI may lead to decreases in extreme rainfall events from ARs affecting western North America and increases in moderate rainfall events by the end of the century (Shields et al., 2022). However, to date, little research has focused on the impact of SAI on the future changes in weather systems over East Asia and the difference between G6sulfur and G6solar simulations which can provide more insights into underlying mechanisms of SAI.



This study aims to assess the possible impact of SAI geoengineering on the different characteristics of ARs, including their frequency, size and precipitation, in the populated regions of East Asia (e.g. southeastern China, Japan and the Korean Peninsula) by using an ensemble of the G6sulfur and G6solar simulations from a state-of-the-art GCM. We also investigate the effect of SAI on the large-scale circulations related to AR activity to understand the environmental mechanisms driving the changes in ARs in East Asia. This study helps to achieve a better understanding of the AR-aerosol interactions and inform the outcome of SAI in terms of the changes in mean and extreme precipitation from ARs. Following this section, Section 2 describes the climate data from the GCM and climate reanalysis datasets and the identification method for detecting AR features in the used data. Section 3 presents the results and Section 4 summarises and discusses the findings.

## 2 Experiments, data, and methods

### 2.1 The G6 experiments based on UKESM1

In this study, the G6sulfur and G6solar experiments in the sixth phase of GeoMIP are applied for analysing the AR characteristics under SAI. The G6sulfur experiment simulates future climate under the SSP5-8.5 scenario with stratospheric $SO_2$ injection aiming to reduce the global mean air temperature to the level under the SSP2-4.5 scenario (Kravitz et al., 2015; Jones et al., 2021; 2022). The SAI is continuous and applied between $10°N$ and $10°S$ along the Greenwich meridian at 18–20 km altitude. The injection rate is adjusted every 10 years to achieve a simulated decadal global-mean temperature to within ±0.2 K of that under the SSP2-4.5 medium forcing scenario. The G6solar experiment has the same goal but achieves it by the idealized gradual reduction of the solar constant (Kravitz et al., 2015; Visioni et al., 2022). Although such a solar dimming scenario is highly idealized compared to the SAI strategy prescribed by G6sulfur, it provides an important reference of a cooled future climate excluding the SAI-related aerosol-climate interaction; hence, its comparison with G6sulfur helps understand the effect of SAI.

The G6 experiment outputs are from the UK Earth System Model (UKESM1, Sellar et al., 2019), a contributing model to the current Coupled Model Intercomparison Project Phase 6 (CMIP6, Eyring et al., 2016). The atmospheric component of UKESM1 is at a spatial resolution of 1.875°×1.25° and 85 hybrid levels extending up to 85-km. The model applies the ENDGame dynamical core, a semi-implicit semi-Lagrangian scheme to solve the non-hydrostatic, fully compressible Navier-Stokes equations (Wood et al., 2014; Mulcahy et al., 2018; Walters et al., 2017). The atmospheric component is coupled to the NEMO (Nucleus for European Modelling of the Ocean) vn3.6 ocean model with spatial resolution of ~1° and 75 levels (Storkey et al., 2018). The land component applies the Joint UK Land Environment Simulator (Best et al., 2011). The atmospheric chemistry model in UKESM1 applies the United Kingdom Chemistry and Aerosols modelling framework and considers a Stratospheric-Tropospheric scheme with aerosol chemistry and online photolysis (Morgenstern et al., 2009; Mann et al., 2010; Archibald et al., 2020). The model also simulates the marine carbon cycle using a biogeochemical model described by Yool et al. (2013). For the experiments considered in this study, UKESM1 is run with three ensemble members





(identifier index: r1i1p1f2, r4i1p1f2 and r8i1p1f2) corresponding to different realization setups. The future changes in AR
characteristics are analysed by calculating the ensemble-mean differences between those identified in the future climate
simulations (SSP5-8.5, SSP2-4.5, G6sulfur and G6solar) for the period 2071-2100 and those identified during the baseline
period 1981-2010 using the UKESM1 historical simulations for CMIP6. An evaluation of the performance of UKESM1 in
simulating the climatology of the detected ARs and associated environmental fields and precipitation is given in the
Supplementary Information. Although some biases are noted, UKESM1 displays some consistency of the diagnosed AR
properties with those identified in the historical climate reanalysis ERA5 (Hersbach et al., 2020) and observed precipitation
(Hamada et al., 2011; Yatagai et al., 2014). For the ability to capture the modulation of ARs by the large-scale circulation,
particularly the East Asian Jet Stream (EAJS), comparisons of the correlation between AR frequency and the jet intensity in
terms of the seasonal mean EAJS index (EAJSI) are made between ERA5 and UKESM1 in the Supplementary Information.
It is found that UKESM1 reasonably captures the negative correlations between AR frequency to the north of 35°N and the
strength of westerly jet stream in terms of the East Asian Jet Stream Index (EAJSI, Lu et al., 2011) compared to the ERA5
reanalysis dataset (Figure S4a, b). Hence, the model can provide useful insight into the impact of SAI on ARs and its
environmental drivers. In addition, at present UKESM1 is the only model that provides outputs of 6-hourly pressure-level
winds and specific humidity data that satisfy the requirement of the used ARDT; hence, the UKESM1 simulations are
focused in this research and the multi-GCM ensembles conducted by the G6 experiments of GeoMIP (e.g. Jones et al., 2021;
2022; Visioni et al., 2021) are not used.

## 2.2 AR Identification

The analyses of AR characteristics in gridded climate data rely on the Atmospheric River Detection Tools (ARDTs).
According to the current ARTMIP project (Shields et al., 2018; Rutz et al., 2019; O'Brien et al., 2022), the diversity of
ARDTs is a major cause of uncertainties in the diagnoses of AR metrics; hence, the use of ARDT should be carefully
selected and tuned for the specific research objectives. This study is based on the ARDT developed by Liang and Yong
(2022) for the objective AR identification in the current UK Met Office GCM. This method detects ARs by isolating
relatively strong low-tropospheric moisture transports from their large-scale background. The identification procedure
comprises three steps, i.e. isolation of continuous high regions of vertically Integrated Water Vapor Transport (IVT),
geometric analysis and diagnostic field computations. Specific procedures of the chosen ARDT, including the calculations of
6-hourly IVT, geometry criteria, filtering of tropical moisture filaments and the diagnoses of AR-associated precipitation, are
described in Liang et al. (2022) and Liang and Yong (2022), while here a different thresholding method is used for the
isolation of continuous AR plumes in gridded climate data. First, following the methodology of ARDTs used in East Asia
(e.g. Pan and Lu, 2019; 2020; Park et al., 2021; Kim et al., 2021), the updated algorithm uses spatially varying thresholds of
IVT in terms of the 85[th] percentile of IVT for each month of the targeted 30-year periods. Second, as per Pan and Lu (2019;
2020), the Gaussian filter with a bandwidth of 6° is applied to the IVT threshold fields. The lower limit of the thresholds (i.e.



the lowest boundary value of the isolated AR plumes) is determined by the 80$^{th}$ percentile of IVT over the region from 40$^{\circ}$E to 120$^{\circ}$W, 20$^{\circ}$S to 60$^{\circ}$N. These help to discern AR plumes from the large-scale background with relatively weak IVT and avoid the influence of high-frequency noise in IVT fields so that more coherent features can be obtained. In addition, in

contrast to the use of a fixed threshold, the relative thresholding methods scale the thresholds respectively for the historical and future climate simulations, which helps remove influences of the large-scale thermodynamic factors that are solely governed by the Clausius-Clapeyron relationship so that the dynamic changes in ARs due to different external climate forcings are focused. This ARDT setup has displayed reliability in detecting ARs over East Asia and exhibited some similarities to those in the recent ARTMIP protocol, which are displayed in the Supplementary Information of Liang et al.

170    (2022).

## 3 Results

### 3.1 Changes in AR-associated environments

Before analysing the impacts of SAI on the identified AR characteristics, such as the AR frequency, plume geometries and

AR-related precipitation, we first assess the changes in AR-associated large-scale environmental fields and discuss their potential influences on ARs. Figure 1 shows the ensemble mean of absolute changes in the low-level large-scale averaged over the main AR season (May to September, MJJAS) for the future period (2071-2100) relative to the historical baseline period (1981-2010). The mean 850-hPa geopotential height during the historical baseline period shows the domination of the Western Pacific Subtropical High (WPSH), an important monsoon system that determines the spatial distribution of ARs

(Pan and Lu, 2020; Park et al., 2021). Simulations under the SSP5-8.5 scenario project a meridional expansion of WPSH with significant increases in geopotential height to the south of 20°N (maximum change centred around 20°N, 85°E coloured in Figure 1a). This drives an intensification of the southwest monsoon across about 20°N and leads to stronger stationary moisture transport from the moisture source region over the Bay of Bengal. Also, a stronger low-level convergence is seen across 30-45°N, which may facilitate the geneses of cyclonic transient eddies that are associated with pole-ward moisture

transport. These changes imply a more favourable environment for AR activities in the study region. Similar patterns are noted under SSP2-4.5 (Figure 1b) while the magnitude of increases in the 850-hPa geopotential height is reduced by about one-third compared to SSP5-8.5. Both G6sulfur (Figure 1c) and G6solar (Figure 1d) show ameliorated changes with respect to SSP5-8.5. Although the SSP2-4.5, G6sulfur and G6solar experiments present similar magnitudes of changes, there are noticeable differences that are investigated further.


Figure 2 shows comparisons of the simulated future low-level circulations between G6sulfur and other experiments. Compared with SSP5-8.5 (Figure 2a), the simulated SAI by G6sulfur leads to smaller increases in the geopotential height at 850-hPa over most of the study region despite the North Pacific (near 40°N, 165°E). This leads to an anticyclonic anomaly,





in terms of the 850-hPa wind field, that favours the northeastward moisture transport of ARs across 40-55°N. Compared with SSP2-4.5 (Figure 2a), G6sulfur shows negative anomalies of the geopotential height over the continent and positive anomalies over the ocean. This implies a more pronounced land-sea thermal contrast that drives an intensification of the southwesterly monsoon flow and the related cyclonic shear across the upstream of the main AR-active region (25-30°N, 105-120°E). Similar patterns of circulation differences are presented when comparing G6sulfur with G6solar (Figure 2c). Therefore, although G6sulfur simulates ameliorated changes in the low-level circulation compared to SSP5-8.5, the low-level environments are more favourable for AR activity for G6sulfur compared to SSP2-4.5 and G6solar. Thus, although the simulated SAI strategy is successful in reducing many of the changes apparent under SSP5-8.5, some differences are evident between G6sulfur, G6solar, and SSP2-4.5 (which all have the same global mean temperature) which is due to the existence of the aerosol-climate feedback under SAI.

The meridional displacement of ARs is related to various high-tropospheric systems including the westerly jet streams and the associated upper-level high pressure (Payne and Magnusdottir 2015; Liang et al., 2022). Following the analyses of Kamae et al. (2017) and Liang et al. (2022), Figure 3 assesses changes in the AR-related upper-level environments including the EAJS in terms of the 200-hPa wind field and the upper-level thermal expansion in terms of 200-500-hPa geopotential thickness. For the warming scenario under SSP5-8.5 (Figure 3a), significant increases (p-value < 0.05) in the thermal expansion are observed over the study region with a maximum increase of more than 230 gpm near northeastern China (130°E, 50°N) and to the south of the WPSH centre (170°E, 20°N). This indicates a pronounced expansion of the South Asian High. The increased thermal expansion to the north of 45°N leads to a decreased meridional thickness gradient and consequently drives a significant weakening (by up to 4 m s⁻¹) of the EAJS across 30-45°N. The projected weakening of the EAJS favours a northward shift of ARs according to the correlation analyses by Liang et al. (2022) and those discussed later (Figure S4, discussed in Section 3.2). Smaller magnitudes of changes (by about 90 gpm) in the geopotential thickness are shown over most of the region under SSP2-4.5 (Figure 3b) and this also results in less apparent weakening of the EAJS. However, G6sulfur projects the most pronounced weakening of EAJS (by up to 8 m s-1) due to the significant decrease in the thickness gradient between 30-45°N, which is associated with the reduced increases in the geopotential thickness to the south of 30°N with magnitudes greater than any other experiments. The further weakening of the EAJS is also related to the presence of the increased maximum to the north (near 130°E, 50°N) though the magnitude is smaller (by about 70 gpm) than that in SSP5-8.5. In contrast, G6solar (Figure 3d) projects similar patterns and magnitudes of the changes in the upper-level thermal expansion and winds compared to SSP2-4.5. To further present the effect of SAI on the AR-related upper-level environments, the absolute differences of 200-500-hPa geopotential thickness and 200-hPa winds between G6sulfur and other future climate simulations are displayed (Figure 4). The comparisons confirm that SAI can lead to further weakening of the EAJS in a warmer climate even though it effectively ameliorates the increase in the upper-level thermal expansion of the high-emission scenario (shown by the negative red contours in Figure 4a). Compared with SSP2-4.5 and G6solar, the



injected sulfur dioxide simulated by G6sulfur leads to further decreases in the low-latitude thickness and positive anomalies to the north of 40°N (Figure 4b, c).

In summary, the UKESM1 simulations under the SSP5-8.5 high-emission scenario project a strengthening of the upstream monsoon flows and the downstream convergence that are favourable for AR activity in most of East Asia. In addition, a weakening of the EAJS driven by the increased upper-level thermal expansion at high latitudes is displayed. The experiments SSP2-4.5, G6sulfur and G6solar show ameliorated changes in the low-level circulation associated with ARs; however, compared to SSP2-4.5 and G6solar, the simulated deployment of SAI in G6sulfur exacerbates the weakening of the
upper-level westerly jet, which are linked to the cooling effect of the injected scattering aerosols which is more concentrated across the lower latitudes. This is evident through the comparisons between G6sulfur and G6solar in terms of the global distribution of aerosol optical depth determined at 550 nm and the surface air temperature according to the study of Jones et al. (2021).

**3.2 Changes in AR features**

Now we analyse the future changes in AR features, including AR frequency, size, wind speeds and moisture content, in a warmer climate and that affected by SAI. Under the SSP5-8.5 scenario (Figure 5a), the ensemble mean of UKESM1 projects a significant increase (p-value < 0.05) in AR frequency over most of southern and eastern China, the Korean Peninsula and Japan. The greatest magnitude of increase (by above 0.8%) is seen in southern China. Some decreases by up to 0.1% are
noted over northeastern China. Projected decreases in frequency are shown to the south of 20°N, which is linked to the domination of the stable high-pressure as the WPSH expands and intensifies (Figure 1a). The significant increase in AR frequency is consistent with the more favourable dynamical condition in the low troposphere as displayed in Figure 1a. Under SSP2-4.5, a similar pattern of frequency change to SSP5-8.5 is projected but with smaller magnitudes (Figure 5b). The experiments G6sulfur (Figure 5c) and G6solar (Figure 5d) show generally ameliorated changes compared to SSP5-8.5.
However, generally any amelioration of AR frequency is the lowest under G6sulfur and G6sulfur shows opposite changes (increase by up to 0.15%) to the north of the main AR-active region, implying a pronounced northward shift of ARs under SAI. This change is linked to the further weakening of the EAJS (Figure 3c) given the negative correlation between AR frequency and EAJSI to the north of 35°N as presented by both the ERA5 reanalysis dataset (Figure S4a) and UKESM1 (Figure S4b); however, underestimation of such a correlation in UKESM1 is noted to the east of 135°E (Figure S4c) and this
bias is partly related to the underestimated AR frequency over the downstream region (see Supplementary Information). Similar negative correlations between upper-level jets and high-latitude AR frequency have also been found by Zhang and Villarini (2018) and Liang et al. (2022). The mechanism behind the enhanced high-latitude ARs by the weakening of EAJS remains elusive, though one of the possible causes is the anomalous low-level convergence to the northwest of the jet core





and divergence to the northeast according to the four-quadrant strait jet model (Uccellini and Johnson, 1979). This
consequently favours the AR-associated northeastward transport of warm moist air across the northern flank of the EAJS.

Figure 6 further illustrates how SAI influences the distribution of AR frequency in terms of differences between G6sulfur
and other experiments. Figure 6a shows that SAI induces significant increases (by up to 0.15%) in AR frequency over
northeastern China though it ameliorates the changes across the south under SSP5-8.5. Compared to SSP2-4.5 and G6solar,
G6sulfur demonstrates limitations in ameliorating the frequency increase over southern China, the Korean Peninsula and
Japan (Figure 6b, c), which is partly linked to the more pronounced intensification of upstream monsoonal flow and low-
level cyclonic shear (Figure 1c).

Figure 7 shows the simulated proportions of AR plumes categorized by different properties for the historical baseline period
(1981-2010) and four experiments of future climate simulation (2070–2099) considering all samples of the three ensemble
members. A prolonged length of ARs is projected under both the SSP4-2.5 and SSP5-8.5 scenarios with increased fractions
for lengths greater than 6000-km and decreased fractions for lengths less than this (Figure 7a), while no apparent change is
noted for AR width (Figure 7b). Figure 7c shows a projected expansion of the size of ARs as fractions increase for ranges of
area greater than $4\times10^6$ km$^2$ and decrease for the lower ranges. This agrees well with the projected size increase of global
ARs by the coupled GCMs of CMIP5/6 (O'Brien et al., 2022). For G6sulfur, some increases in AR width are seen in contrast
to SSP2-4.5 and SSP5-8.5. Also, compared to SSP5-8.5, higher magnitudes of increases in fractions for ARs with sizes
between $3-4\times10^6$ km$^2$ are projected and similar magnitudes of increases are shown for relatively large sizes ($> 4\times10^6$ km$^2$).
The G6solar experiments project similar changes to SSP2-4.5. These changes suggest no apparent effect of SAI in mitigating
the deformation of ARs under climate warming. On the other hand, Figure 7d shows that G6sulfur projects a pronounced
northward shift in ARs with increased fractions of ARs located to the north of 25°N and decreased fractions to the south.
Such a change is not apparent for other experiments. This is corresponding with the increased high-latitude ARs under the
weakening of the EAJS as discussed in Sections 3.1 and 3.2. In Figure 7e, the distribution for SSP5-8.5 shows a weaker wind
speed along AR axes compared to the historical simulations. For the changes in the main tropospheric moisture carried by
ARs, Figure 7f shows a pronounced increase in relatively wet ARs under SSP5-8.5 compared to the historical baseline.
Ameliorated changes in these features are observed for SSP2-4.5, G6sulfur and G6solar, implying that SAI can partly reduce
both the dynamical and thermodynamical responses within AR plumes.

In summary, the SAI strategy simulated by the G6sulfur experiments can partly mitigate the changes in AR features in East
Asia, particularly for the increase in AR frequency over southern China and decreases at low latitudes. It could also partly
ameliorate the changes in the moisture content and low-level wind speeds along AR axes. However, SAI could potentially
induce increases in AR activity across northeastern China, the Korean Peninsula and Japan as it further weakens the EAJS
intensity that is negatively correlated to the high-latitude AR activity. It is also limited to mitigating the changes in AR





geometry under the high-emission scenario. The presented AR-SAI connections could possibly lead to changes in the mean and extreme precipitation of the region and this will be examined in the following section.


## 3.2 Changes in AR-associated precipitation

Previous sections have discussed the impact of SAI on ARs and the associated large-scale environments in East Asia. This section further examines the responses of AR-associated mean and extreme precipitation under the future changes in ARs. For the SSP5-8.5 scenario (2071-2100) relative to the historical baseline period (1981-2010), the ensemble mean of
UKESM1 projects a significant increase (p-value < 0.05) in the annual mean accumulation of AR-associated precipitation across western Japan, the Korean peninsula and most of southern and eastern China with magnitudes of up to above 640 mm per year (Figure 8a). Some decreases are noted in the coastal regions of southern China. Similar changes with smaller magnitudes are seen under the SSP2-4.5 scenario (Figure 8b). Resembling the changes in AR frequency, both G6sulfur and G6solar shows smaller magnitudes of increases across 30°N compared to SSP5-8.5 but greater magnitudes compared to
SSP2-4.5. For the changes in ARs' fractional contribution to annual total precipitation, ARs tend to contribute more precipitation across central and eastern China (by up to 12%) as well as the Korean Peninsula (up to 6%) under SSP5-8.5 (Figure 8e). Decreases in the fraction are seen over northeastern China due to the pronounced increases in non-AR precipitation, which is not the focus of this paper. SSP2-4.5 (Figure 8f) and G6solar (Figure 8h) project less apparent increases in fraction compared to SSP5-8.5. For G6sulfur (Figure 8g), the simulated SAI strategy leads to fewer increases in
fraction across 30°N. In contrast to SSP5-8.5 and SSP2-4.5, significant increases in fraction by up to 4% are seen in northeastern China and by 2-6% are observed in most of Japan.

For the ensemble mean of the simulated heavy rain events associated with ARs (being the number of days accumulated when the daily precipitation amount is greater than 40 mm day$^{-1}$ at a given location and an AR axis is within 350 km), the patterns
of changes shown in Figure 9a-d resemble that for precipitation accumulation. It is noted that G6sulfur (Figure 9c) shows significant increases in AR-associated extreme precipitation in Japan with a magnitude similar to that under SSP5-8.5. Also, G6sulfur generally projects a larger magnitude of increases in the fraction of heavy rain events over central-eastern China and Japan (Figure 9g) compared to other future simulation experiments (Figure 9e, f, h). These changes imply pronounced side effects of SAI on hydrological extremes from ARs in the above-mentioned populated regions.


We now compare the ensemble mean of the AR-associated precipitation patterns between G6sulfur and other experiments. As expected, G6sulfur effectively ameliorates the increase in AR precipitation across 30°N compared to SSP5-8.5 (Figure 10a); however, the simulated SAI strategy significantly exacerbates the increase in AR precipitation across 45°N (p-value < 0.05), which is linked to the increase in high-latitude ARs as shown in Figure 6b and Figure 7d. Figure 10b shows the
comparison between G6sulfur and SSP2-4.5, which indicates a general increase in AR precipitation for most of the study





region, particularly central-northern China (by up to 80 mm per year) and Japan (120 mm per year). Likewise, the comparison between G6sulfur and G6solar indicates a general increase in precipitation (by up to 80 mm per year) for the same regions. For the AR-related heavy rain events, although ameliorated increases are shown in central and eastern China when comparing G6sulfur with SSP5-8.5 (Figure 10d), the simulated SAI strategy exhibits amplified increases in the events

in central-northern China and Japan with respect to SSP2-4.5 (Figure 10e) and G6solar (Figure 10f), implying a considerable side effect of SAI on extreme precipitation.

## 4 Summary and Discussion

In this study, the future changes in ARs and associated precipitation over East Asia by the end of the 21$^{st}$ century are examined using the historical (1981-2010) and future climate (2071-2100) simulations based on UKESM1 under different

external climate forcings. The effect of SAI on ARs is assessed by comparing the experiment G6sulfur with the idealized solar dimming condition (G6solar) and high (SSP5-8.5) and medium (SSP2-4.5) levels of GHG emissions. The conclusions of the paper are summarized as follows:

(1)   Under both the SSP2-4.5 and SSP5-8.5 scenarios, a strengthening of the upstream monsoon flows and the

downstream convergence are projected, which creates a more favourable environment for AR activity in most of East Asia. Consequently, an increase in mid-latitude (near 30°N) AR frequency and AR-associated precipitation relative to the historical baseline period is projected. In addition, a weakening of the EAJS driven by the increase in upper-level thermal expansion at high latitudes is projected. Given the negative correlation between the strength of EAJS and the local AR frequency at higher latitudes, northward shifts in AR activity under the weakening of EAJS

leads to significant increases in AR-associated mean and extreme precipitation at high latitudes, particular northern China and Japan.

(2)   Compared to SSP5-8.5, the experiment G6sulfur simulates ameliorated changes in the low-level environments controlling the activity of ARs by the simulated SAI strategy. The comparison of the identified AR features among the different experiments shows that the simulated SAI strategy is effective in partly mitigating the projected future

increase in AR activity over the study region, particularly southern China, and the increases in the moisture content and low-level wind speeds along AR axes. This implies that both the thermal and thermodynamical responses of ARs can be reduced by the simulated SAI.

(3)   The simulated SAI strategy in G6sulfur exacerbates the weakening of the EAJS due to the concentrated cooling effect of the injected scattering aerosols across the lower latitudes. It also leads to stronger land-sea thermal contrast

that favours ARs with respect to SSP2-4.5 and G6solar. As a result, a side effect of SAI exacerbating the increases in high-latitude AR activity and associated mean and extreme precipitation is observed, particularly over northeastern China, the Korean Peninsula and Japan.



The presented future changes in AR activity over East Asia, particularly the increase in AR frequency over southern China,
agree well with previous AR projection studies (Espinoza et al., 2018; Kamae et al., 2021). The presented increases in AR
length and size under the warming scenarios have also been found globally (Espinoza et al., 2018; Zhao, 2020). These
similarities imply additional confidence in the reported future AR projection in this study. However, any presented changes
should be carefully interpreted considering the bias of UKESM1 in simulating the spatial distributions of AR frequency and
AR precipitation as shown in the Supplementary Information. Particularly, the underestimation of the downstream AR
frequency for UKESM1 compared to the ERA5 reanalysis dataset is possibly related to the use of relatively coarse horizontal
resolution according to Liang and Yong (2022). Nevertheless, the presented future changes in ARs are based on the
ensemble mean of three different realization setups of UKESM1, which is limited to considering model uncertainties. These
should be addressed by future study depending on the potential improvement of the data availability for GeoMIP GCMs,
especially for those run at finer model resolutions. Another limitation is the lack of analyses of the uncertainties associated
with the choice of ARDTs due to the limited resources, although some configuration and the performance of the used ARDT
have demonstrated some similarities compared to other ARDTs in the ARTMIP protocol (Liang et al., 2022). Also, as the
chosen ARDT rescales the IVT thresholds for each month, it partly excludes the signal of AR seasonality thus the projected
change in AR seasonality is not investigated. This should be addressed in future work by changing the current thresholding
setup, such as the use of a 5-month moving time window for rescaling the IVT thresholds (Park et al., 2021).


The side effect of SAI manifested by the thermal response of the upper-tropospheric circulation and its control of the local
AR activity implies that any deployment of SAI should be evaluated with caution given the existence of the
links/teleconnections between the regional climate over the populated regions of East Asia and the large-scale circulation
that is sensitive to the injected aerosol precursor. Another example of the side effect of SAI on precipitation, i.e.
exacerbating precipitation deficit over the Mediterranean, is presented by Jones et al. (2022). Further studies are required to
understand the potential hydrological impacts of SAI. This includes the use of hydrological modelling tools to simulate
hydrological extremes associated with ARs at the watershed level (e.g. Dettinger et al., 2011; Chen et al., 2019) with climate
inputs from the GeoMIP experiments. Furthermore, the presented side effect should be incorporated into dissemination of
climate change information for decision makers involved with adaptation strategies in the populated regions. It also implies
the necessity to optimize the potential deployment of SAI, including adjustments of the injection location and considering
different candidates of the injected material that are more effective in increasing the outgoing radiation with less absorption
(Jones et al., 2016). Overall, this paper calls for better awareness of the consequences brought by any practical deployment
of SAI geoengineering from a perspective of high-impact weather systems and their association with extreme climate events.

**Acknowledgements**



JL and JH would like to acknowledge support from the NERC funded SASSO standard grant (NE/S00212X/1). JH was supported by the Met Office Hadley Centre Climate Programme funded by BEIS. JH would also like to acknowledge support from the NERC funded EXTEND project (NE/W003880/1) and from SilverLining through its Safe Climate Research Initiative. The authors thank Andy Jones for help with the UKESM1 outputs.

## Code/Data availability

All model data used in this work are available from the Earth System Grid Federation (WCRP, 2021; https://esgf-node.llnl.gov/projects/cmip6, last access: 14 July 2021). The used ERA5 reanalysis dataset is downloaded from the Copernicus Climate Data Store. The APHRODITE data is downloaded from its official website managed by the Research Institute for Humanity and Nature.

## Author contribution

JL and JH led the analysis and wrote the paper. Funding acquisition and the computational facility and resources for this research were contributed by JH.

## Competing interests

The authors declare no competing interest.

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





**Figure 1: Absolute changes in MJJAS-mean geopotential height (shaded, unit: geopotential meter / gpm) and winds (vectors) at 850-hPa for the future period of 2071-2100 relative to the historical baseline during 1981-2010. Black contours show the MJJAS-mean 850-hPa geopotential height during the historical baseline period. Areas with a surface pressure below the 850 hPa level are shown in white.**



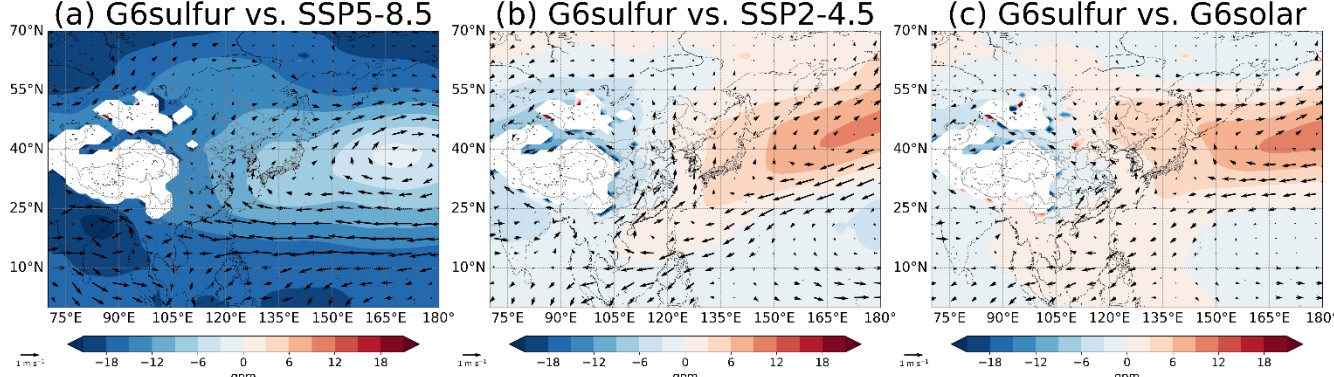

**Figure 2: Differences in MJJAS-mean geopotential height (shaded) and winds (vectors) at 850-hPa for the future period of 2071-2100 between G6sulfur and SSP5-8.5 (a), SSP2-4.5 (b) and G6solar (c).**



**Figure 3: Absolute changes in MJJAS-mean geopotential thickness between 200 and 500-hPa (red contours, unit: gpm) and 200-hPa wind speeds (shaded) for the future period of 2071-2100 relative to the historical baseline during 1981-2010. Black contours (unit: gpm) show the MJJAS-mean 200-500-hPa geopotential thickness and vectors show the MJJAS-mean 200-hPa wind fields during the historical baseline period.**





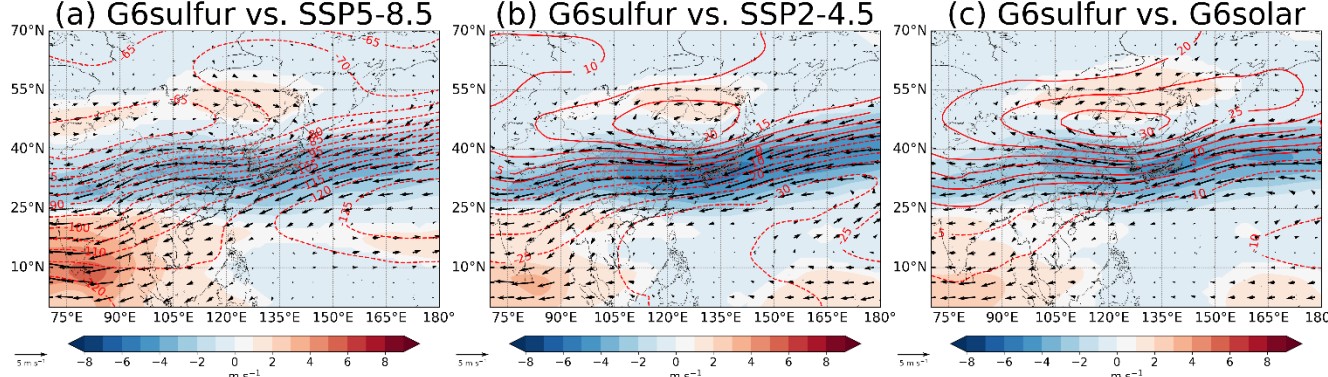

**Figure 4: Differences in MJJAS-mean geopotential thickness between 200 and 500-hPa (red contours) and 200-hPa winds (shaded for velocity and vectors for direction) for the future period of 2071-2100 between G6sulfur and SSP5-8.5 (a), SSP2-4.5 (b) and**
**G6solar (c).**





**Figure 5: Absolute changes (shaded) in annual mean AR frequency (fraction of 6-hourly time steps) for the future period of 2071-2100 relative to the historical baseline during 1981-2010. Black contours show the annual mean AR frequency during the historical baseline period. Stippling indicates changes that are statistically significant at a confidence level of > 95% (p-value < 0.05 based on the Student's t-test).**



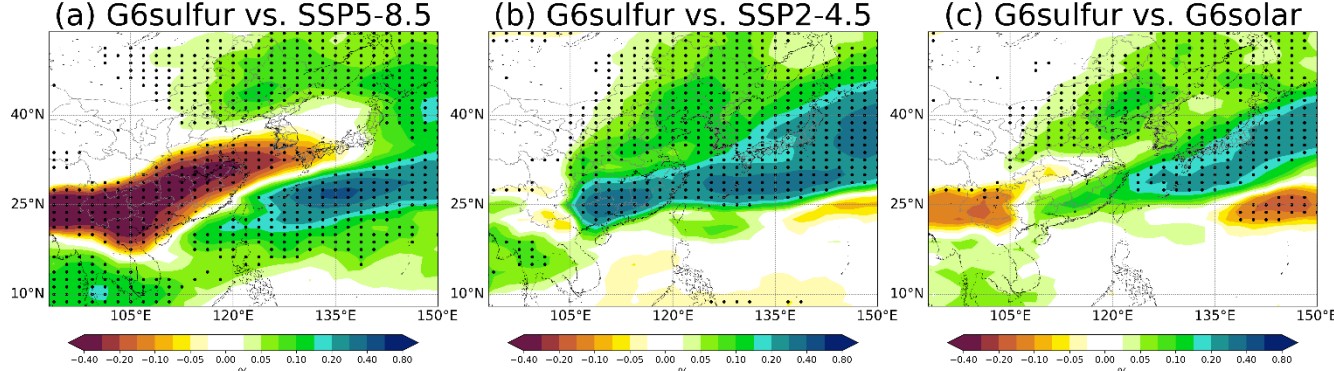

**Figure 6: Differences in annual mean AR frequency for the future period of 2071-2100 between G6sulfur and SSP5-8.5 (a), SSP2-4.5 (b) and G6solar (c). Stippling indicates differences that are statistically significant at a confidence level of > 95% (p-value < 0.05 based on the Student's t-test).**





**Figure 7: Box-and-whisker plots for fractions of identified AR plumes with centroids within 80-160°E and 10-60°N categorized by different metrics, including AR length (a), width (b), size (area of plume coverage, c), mean latitude of AR axis (d), wind speed vertically averaged over 925-700-hPa (e) and the integrated water vapour content (IWV) in the main troposphere (f). From low to high, each box shows the minimum, the first quartile (25th percentile), the median, the third quartile (75th percentile) and the maximum of the fractions.**





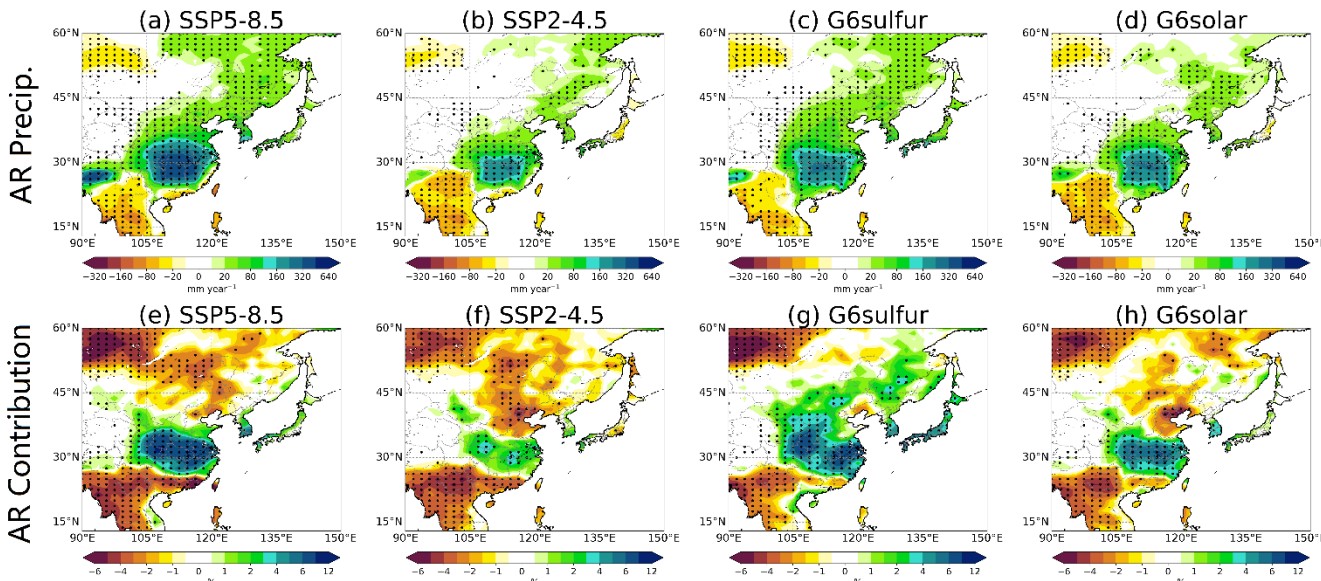

**Figure 8: Absolute changes in the annual accumulation of AR-related precipitation (a-d) and the fractional contribution of ARs to annual total precipitation amount (e-h) for the future period (2071-2100) relative to the historical baseline (1981-2010). Stippling indicates changes that are statistically significant at a confidence level of > 95% (p-value < 0.05 based on the Student's t-test).**

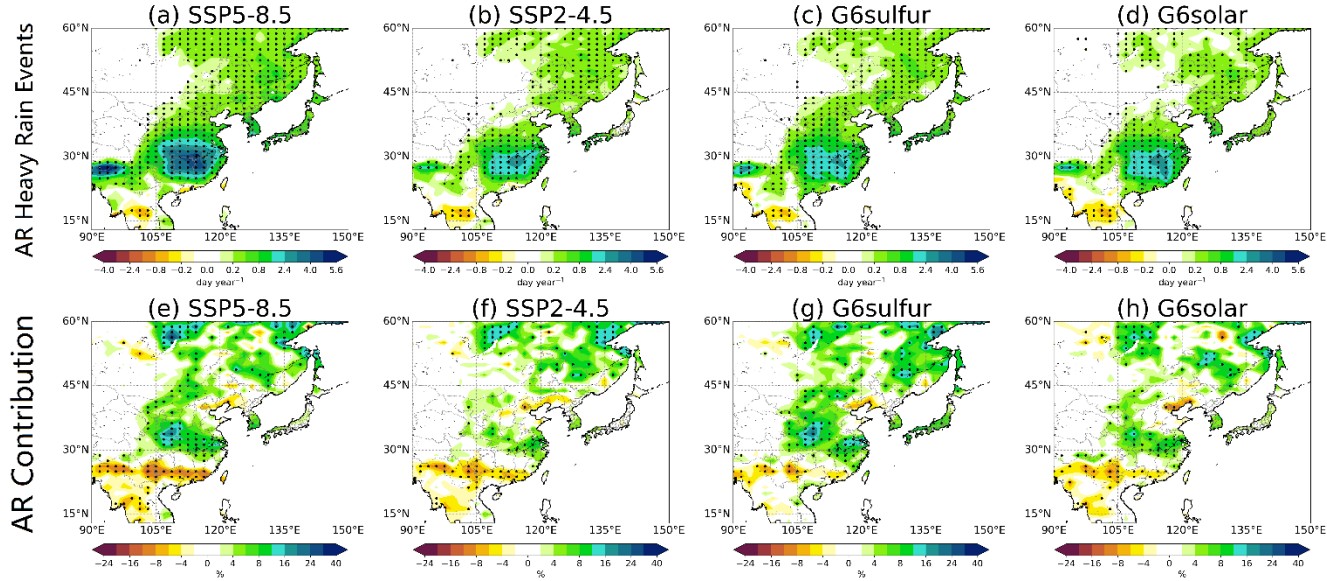

**Figure 9: As Figure 8, but for the occurrence of AR-related events (a-d) of heavy rain (daily precipitation > 40 mm) and the contribution of ARs to the annual total heavy rain events (e-h).**



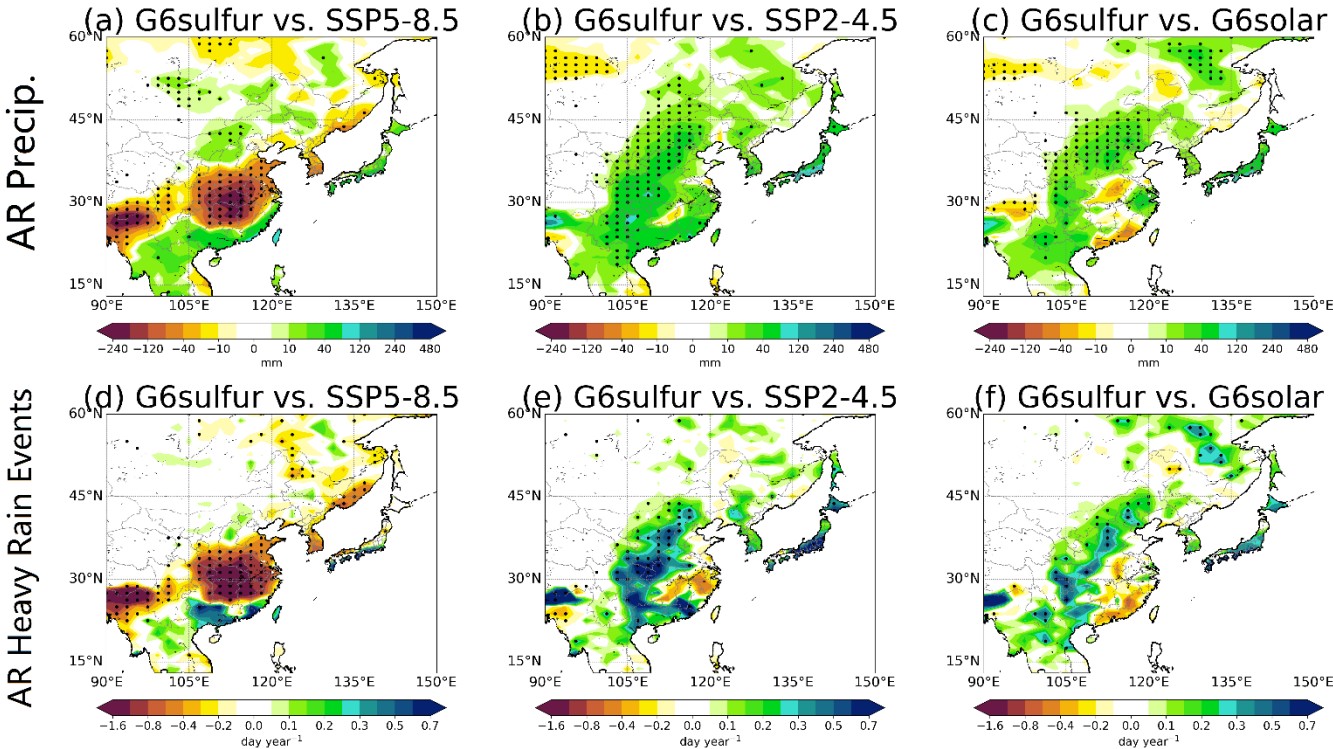

**Figure 10: Differences in annual accumulation of AR-related precipitation (a-c) and heavy rain events (d-f) for the future period of 2071-2100 between G6sulfur and SSP5-8.5 (a, d), SSP2-4.5 (b, e) and G6solar (c, f). Stippling indicates differences that are statistically significant at a confidence level > 95% (p-value < 0.05 based on the Student's t-test).**