# Peer review of "Future changes in atmospheric rivers over East Asia under stratospheric aerosol intervention"

_EGUsphere, 2022_

## Referee Comment (RC2)

Review for EGUsphere

Title: Future changes in atmospheric rivers over East Asia under stratospheric aerosol intervention

Authors: Ju Liang, Jim Haywood

Liang and Haywood analyze a series of climate change and geoengineering simulations from the UK ESM to diagnose impacts on atmospheric rivers (ARs) in the East Asia region. Both SAI and solar dimming experiments are used and compared to SSP5-8.5 and SSP2-4.5. The paper looks at both dynamical differences, precipitation impacts, and geometric AR structure. They find that ARs increase in frequency with related precipitation in warming scenarios, especially for southern China. SAI implementation reduces these impacts over the lower mid-latitudes and subtropics, but potentially exacerbates AR activity and impacts over upper midlatitudes, essentially due to dynamical shifts.

This paper is very well written and organized with clear figures and reasoning. The dynamical arguments are robust regarding connections to ARs. My main issue is with the AR metrics/features section, specifically inferences that are made in the text that are based on a box and whisker figure that does not necessarily support the conclusions stated. The differences between most of the simulation types are small with the spread often overlapping, and all based on a limited ensemble size. I recommend presenting this information in a more convincing manner, or removing this figure. Given only 1 ARDT is used, more information should be provided (see line comments below). Also, more information on UK ESM is needed, specifically for global, climate base states for SSPs compared to Geo simulations (again, see line comments below).

Lines 44-46: ARTMIP Tier 2, and especially O'Brien et al. 2022, have robustly shown that ARDT uncertainty far outweighs model uncertainty. It is also true that ARDT uncertainty outweighs uncertainty across Reanalysis products as well. The current sentence doesn't really communicate this finding, please adjust.
 https://agupubs.onlinelibrary.wiley.com/doi/10.1029/2021JD036155.

Line 52: Did you mean this reference? I don't see it in the reference list:
https://doi.org/10.1029/2019GL085565.

Line 110: What is the impact of adjusting the injection period? Also, the injection site? You touch on this in the discussion, but I have these additional questions). I realize that many models use a controller that bases locations based on temperature gradients which is determined by the climate base state. Is this true here as well? My understanding is that injection sites can make a big difference in terms of where aerosols circulate, and thus climate. Maybe a sentence or two acknowledging uncertainty due to injection site (or computation of injection site) can be added.

Line 112: Just to confirm, G6Solar, is dimming imposed on the base state of SSP5-8.5?

Line 120: The relatively low resolution used here compared to many AR studies should be addressed. For synoptics, this resolution is absolutely fine, but for regional precipitation, maybe

not, as seen in your supplemental Figure S2. Therefore, these limitations need to be discussed in the main paper.

Line 129: Why only 3 members? This seems a bit small to capture the internal variability, which is the purpose of ensembles.

Lines 134-145: Thank you for this supplemental information on model biases as they pertain to ARs! Other model biases important for SAI should also be included in supplemental, as well as some base state information to provide context as to how UKESM1 performs with and without SAI. For those not regularly following GeoMIP and SAI literature, this would be very helpful information. (Maybe global maps to see impact on other areas)? Also, in Figure S1, are the gray regions those with PS below 850mb?

Line 131: AR tracking was performed for each ensemble member, then the means were computed, correct?

Lines 150-170, AR detection: Thank you for this description. I did go to the supplemental material of Liang et al. 2022. I am assuming that the ARDT used in this work is the same ARIA-Asia described in this paper? If not, what are the differences and how do they compare to Figure S1 in that paper, i.e. other ARTMIP ARDTs? I would highly recommend a version of that figure to be included in the supplemental material here given it is directly relevant. Because not all readers have the same level of accessibility to papers,  it is best to have everything in one spot, especially when that information is so closely tied to the interpretation of results. Given this, I also recommend a more detailed description (including geometry, see Figure 7 comments) should be provided in the supplemental rather than relying on the Liang references.

Figure 1: Here and elsewhere, please darken the continental outlines, they are really hard to see.

Line 193: Seems like there are some words missing at the end of the sentence?

Line 197: Is there more monsoonal precip here as well? The different SSP responses are interesting! For the aerosol feedbacks, is there a reference (such as the Jones et al 2021 you use later), or is this current work?

Line 205: Shields et al., 2022  also shows that the dynamical AR climate/SAI change response is dictated by the jets.

Line 219: with magnitudes (of weakening) greater than any other experiment?

Figures 5,6,7:  Unit should be labeled as (fraction of 6-hourly time steps) and not %

Line 256/257/324/345/356 and maybe elsewhere): Not sure "high latitude" is appropriate here, perhaps, upper-midlatitudes.

Line 261: I am confused on the unit… % of time period, or fraction of time period?  Percent is usually given in units 0-100%, whereas fraction is 0 to 1.

Figure 7 and discussion:

- Are some of the geometrical qualities somewhat "baked into" the ARDT by definition (line 155)? The geometrical conditions are not specified in the AR detection section, but given this figure, I think they should be stated. Perhaps more detail in the ARDT in the supplemental material is needed, rather than relying on the given references.

- Although the discussion (I think) is using mostly the mean values and movement of the max and min whiskers, I am a little uncomfortable with these statements given that the *range* of possibilities within each category are quite similar for all panels except 7f. Some of the metrics have increased, or decreased, *range envelopes*, but that is about all you can say, especially given the small sample/ ensemble size. Please rewrite this section keeping in mind the uncertainties of each metric. The fact that many of these whiskers (for each metric) completely overlap each other means that there is high uncertainty.

Line 287-288: AR frequency and size (as shown by the O'Brien paper) are very much tied to ARDT specifications. This, plus Fig 7, does not convince this statement can be made for anything over than moisture content for certain experiments.

Line 292-293: Same as above

Figure 9 and 10: Because this area also experiences tropical cyclones, again, a brief sentence on how TCs are filtered out of the ARDT would be useful in the AR detection section (or supplemental). This may be for a different study, but it would be interesting to understand how much of the total precipitation is TC-related vs AR-related.

Line 351-352: An increase in wind speed along the AR core is not supported by Figure 7.

---

## Author Comment (AC1)

Dear reviewer,

We would like to express our sincere appreciation for your comprehensive reviews and comments that greatly help us to improve our manuscript. This manuscript has been thoroughly revised based on your comments. The revisions we make based on each comment is explained point-by-point.

1) **It seems to me that discussion on physical reasons for changes in upper tropospheric thickness, atmospheric circulation, and AR activity are not sufficient. Previous studies identified that forcing factors and ocean warming result in different patterns of changes in thickness and atmospheric circulation over East Asia (e.g. doi:10.1007/s00382-014-2073-0, doi:10.1007/s00382-014-2146-0, doi:10.1038/ngeo2449, doi:10.2151/sola.2018-010). Such discussion, especially in terms of land-sea warming contrast, should be added to help better understanding on physical mechanisms of different changes in upper troposphere in response to different forcing factors.**

Authors:

We thank the reviewer for the criticism on the discussion of the environmental factors associated with changes in ARs. In response to this, additional figures (Figure 3 and Figure S5) and more discussion on the changes in the land-sea thermal contrast in terms of the absolute changes in surface temperature have been added to the main text and the supplementary information. We note that the changes in surface temperature provide additional evidence of the increased land-sea thermal contrast in the projected warmer climate and the associated increases in AR activity over East Asia. As discussed in lines 216-219, some patterns of changes in the land-sea thermal contrast, tropospheric thermal expansion and EAJS exhibit similarities to the studies of Kamae et al. (2014) and Endo et al. (2018) and we thank the reviewer for recommending these helpful references. We have also discussed that the mid-latitude ocean warming under the simulated SAI may lead to reversed "cold ocean/warm land" (COWL) pattern that enhances atmospheric blocking (Mullen 1989), which partly explain the increase in the mid-latitude AR activity (Pohl et al., 2021; discussed in lines 199-201).

Unfortunately, the GEOMIP protocol does not provide AMIP-style simulations for presenting the contrasting patterns of environmental responses under the radiative forcing of $CO_2$ increases compared to those under only the warming of the SSTs (Kamae et al., 2014; Shaw and Voigt, 2015). We acknowledge that such an important mechanism should be supplemented in future works on geoengineering simulations (lines 356-360).

2) **I also recommend to include global maps of thickness, geopotential, and wind patterns in response to SSP, sulfur, and solar forcing. Such global figures may help more reasonable understanding on changes in atmospheric river activity over East Asia.**

Authors:

We agree with the reviewer that a bigger picture of the AR-related large-scale environments is needed. In response to this, we have enlarged some of the maps to include not only East Asia but also most of the North Pacific, Bay of Bengal and the East Indian Ocean. However, presenting the global distributions of fields may include too many irrelevant systems (e.g. the jet stream across the Roaring Forties and Furious Fifties of the southern hemisphere) – as the title says, we are focusing on East Asia, not on the globe as a whole. We also do not include too much information on the monsoonal flows over South Asia and the upstream Somalia region as ARs and the associated environments over these regions are out of the scope of this study.

3) **Line 211: I don't think the results of these simulations support the SAH expansion. Increasing geopotential height are found outside of SAH region. The increase in geopotential height at 40-60N doesn't indicate SAH expansion but enhanced land-sea contrast over East Asia.**

Authors:
In response to the comment, this statement has been removed so that the discussion on the land-sea contrast is better focused in this paragraph.

4) **I also think more discussion on future changes in summertime storm track is needed for better understanding on changes in AR activity. Previous studies pointed out shift and weakening of storm track over the summertime North Pacific (doi:10.1029/2020JD032701, doi:10.1007/s00704-008-0083-8). Such changes in storm track and associated changes in jet stream should be investigated in more detail, because such mid-latitude disturbances should be essentially important for AR activity.**

Authors:
Following the comment, in lines 235-244 and Figures 6 and S6 we have made additional analyses of the summer storm tracks using the root-mean-square field of MSLP bandpass-filtered by the Laczos resampling method according to one the suggested reference (Harvey et al., 2020). Please note that, although ARs have been frequently observed to be related to extratropical storms, the storm track does not provide good explanation for the changes in the semi-stationary AR flows over the upstream region where the societal impact of ARs is apparent (lines 287-290).

5) **In figures 9 and 10, the authors investigated changes in heavy rainfall associated with ARs. The results are very interesting. However, when you discuss fractional contribution of ARs to heavy rainfall, you need to discuss effect of tropical cyclones. As indicated in many previous studies (e.g. doi:10.2151/sola.2017-002), future changes in tropical cyclone frequency/intensity are primarily important for future changes in heavy rainfall over the western North Pacific. Therefore, discussion on AR frequency itself is not sufficient for discussion on fractional contribution to heavy rainfall.**

**I recommend the authors to add discussion on future changes in TC-related heavy rainfall over this region in these simulations.**

Authors:
We acknowledge that tropical cyclones (TCs) can bring destructive winds and extreme precipitation to the coastal communities and their comparison with ARs is useful to understand the roles of different high-impact synoptic systems in regional climate. However, we decided not to include analyses of TC precipitation as this study is based on a GCM that has a relatively coarse resolution in which TCs are not well resolved. Diagnoses of TCs in such models usually rely on empirical indices that quantify the genesis density of TCs by considering the large-scale thermodynamical factors (e.g. Webster et al., 2005; Jones et al., 2017; Emanuel 2021) but such a method is unable to examine TC-related precipitation.

Thank you for your consideration of publication.

Sincerely yours,
Ju Liang (first author, corresponding author)
and
Jim Haywood

**Reference:**

➤ Emanuel, K., 2005. Increasing destructiveness of tropical cyclones over the past 30 years. Nature 436, 686–8. https://doi.org/10.1038/nature03906

➤ Endo, H., Kitoh, A., Ueda, H., 2018. A unique feature of the Asian summer monsoon response to global warming: The role of different land-sea thermal contrast change between the lower and upper troposphere. Sci. Online Lett. Atmos. 14, 57–63. https://doi.org/10.2151/SOLA.2018-010

➤ Harvey, B.J., Cook, P., Shaffrey, L.C., Schiemann, R., 2020. The Response of the Northern Hemisphere Storm Tracks and Jet Streams to Climate Change in the CMIP3, CMIP5, and CMIP6 Climate Models. J. Geophys. Res. Atmos. 125, e2020JD032701. https://doi.org/10.1029/2020JD032701

➤ Jones, A.C., Haywood, J.M., Dunstone, N., Emanuel, K., Hawcroft, M.K., Hodges, K.I., Jones, A., 2017. Impacts of hemispheric solar geoengineering on tropical cyclone frequency. Nat. Commun. 2017 81 8, 1–10. https://doi.org/10.1038/s41467-017-01606-0

➤ Kamae, Y., Watanabe, M., Kimoto, M., Shiogama, H., 2014. Summertime land–sea thermal contrast and atmospheric circulation over East Asia in a warming climate—Part II: Importance of CO2-induced continental warming. Clim. Dyn. 43, 2569–2583. https://doi.org/10.1007/s00382-014-2146-0

➢ Mullen, S.L., 1989. Model Experiments on the Impact of Pacific Sea Surface Temperature Anomalies on Blocking Frequency. J. Clim. 2, 997–1013. https://doi.org/10.1175/1520-0442(1989)002<0997:meotio>2.0.co;2

➢ Pohl, B., Favier, V., Wille, J., Udy, D.G., Vance, T.R., Pergaud, J., Dutrievoz, N., Blanchet, J., Kittel, C., Amory, C., Krinner, G., Codron, F., 2021. Relationship Between Weather Regimes and Atmospheric Rivers in East Antarctica. J. Geophys. Res. Atmos. 126, e2021JD035294. https://doi.org/10.1029/2021JD035294

➢ Voigt, A., Shaw, T.A., 2015. Circulation response to warming shaped by radiative changes of clouds and water vapour. Nat. Geosci. 8, 102–106. https://doi.org/10.1038/ngeo2345

➢ Webster, P.J., Holland, G.J., Curry, J.A., Chang, H.-R., 2005. Changes in tropical cyclone number, duration, and intensity in a warming environment. Science 309, 1844–6. https://doi.org/10.1126/science.1116448

---

## Author Comment (AC2)

Dear reviewer,

We would like to express our sincere appreciation for your detailed comments that greatly help us to improve our manuscript. This manuscript has been thoroughly revised based on your comments. The revisions we make based on each comment is explained point-by-point.

1) **My main issue is with the AR metrics/features section, specifically inferences that are made in the text that are based on a box and whisker figure that does not necessarily support the conclusions stated. The differences between most of the simulation types are small with the spread often overlapping, and all based on a limited ensemble size. I recommend presenting this information in a more convincing manner, or removing this figure. Given only 1 ARDT is used, more information should be provided (see line comments below). Also, more information on UK ESM is needed, specifically for global, climate base states for SSPs compared to Geo simulations (again, see line comments below).**

Authors:
As suggested, we have removed the box and whisker figure as this provides limited information on the statistical significance of the changes in AR features. This also considers the reviewer's comment on the uncertainty of ARDT in diagnosing AR axes and sizes (see reply to comment 19).

We have also considered the reviewer's suggestion on the need for more evaluation of the performance of UKESM1 in simulating the large-scale environments associated with SAI (see reply to comment 8).

2) **Lines 44-46: ARTMIP Tier 2, and especially O'Brien et al. 2022, have robustly shown that ARDT uncertainty far outweighs model uncertainty. It is also true that ARDT uncertainty outweighs uncertainty across Reanalysis products as well. The current sentence doesn't really communicate this finding, please adjust.**

Authors:
In response to the comment, we have revised the statement as "(ARTMIP, Payne et al., 2020; Collow et al., 2022; O'Brien et al., 2022), have suggested that the projected changes in ARs are susceptible to uncertainties in the algorithms used for detecting ARs and such uncertainties outweigh those in climate models and reanalysis products." (lines 45-47).

3) **Line 52: Did you mean this reference? I don't see it in the reference list: https://doi.org/10.1029/2019GL085565.**

Authors:
We apologize for the mistake and this paper has been added to the reference list.

4) **Line 110: What is the impact of adjusting the injection period? Also, the injection site? You touch on this in the discussion, but I have these additional questions). I realize that many models use a controller that bases locations based on temperature gradients which is determined by the climate base state. Is this true here as well? My understanding is that injection sites can make a big difference in terms of where aerosols circulate, and thus climate. Maybe a sentence or two acknowledging uncertainty due to injection site (or computation of injection site) can be added.**

Authors:

The injection strategy, including the injection period and site, is prescribed in each GCM ensemble member by the G6 experiment of GeoMIP (Kravitz et al., 2015). The GeoMIP G6 protocol does not (yet) explore the uncertainties associated with the choices of injection periods or sites, and specifies injections at the Equator, where the impacts on temperature are expected to be an approximate maximum owing to the dispersion of the aerosol in the Brewer-Dobson circulation. Only two models (CESM and NASA GISS) have currently published results using a control algorithm (MacMartin et al., 2017; Kravitz et al., 2017), although this facility is in development in more models (e.g. UKESM1). In response to the comment, in lines 111-114 we have added brief discussion on the impact of injection site on the meridional thermal contrast and monsoon precipitation.

5) **Line 112: Just to confirm, G6Solar, is dimming imposed on the base state of SSP5-8.5?**

Authors:

That is correct as the description in lines 115-119.

6) **Line 120: The relatively low resolution used here compared to many AR studies should be addressed. For synoptics, this resolution is absolutely fine, but for regional precipitation, maybe not, as seen in your supplemental Figure S2. Therefore, these limitations need to be discussed in the main paper.**

Authors:

The issue related to the relatively coarse model resolution of UKESM1 has been acknowledged and briefly discussed in lines 372-374.

7) **Line 129: Why only 3 members? This seems a bit small to capture the internal variability, which is the purpose of ensembles.**

Authors:

We admit that the ensemble membership is limited, and this is mainly due to the limited computational resources supporting GeoMIP as currently most of the

participating modelling groups set not more than 3 ensemble members for their own simulations (Jones et al., 2022). The setup of the ensemble members for the GeoMIP simulations of UKESM1 also considers the availability of UKESM1's other CMIP6 simulations (i.e. the present-day climate simulations as well as SSP2-4.5 and SSP5-8.5) so that comparisons are allowed.

In response to the comment, the issue of the limited ensemble size is mentioned in lines 374-377.

8) **Lines 134-145: Thank you for this supplemental information on model biases as they pertain to ARs! Other model biases important for SAI should also be included in supplemental, as well as some base state information to provide context as to how UKESM1 performs with and without SAI. For those not regularly following GeoMIP and SAI literature, this would be very helpful information. (Maybe global maps to see impact on other areas)? Also, in Figure S1, are the gray regions those with PS below 850mb?**

Authors:

We agree with the reviewer that the evaluated fields need to be enlarged to better present the biases in UKESM1. Unfortunately, as the ERA5 data we downloaded has been regionally truncated (mainly over the Asian monsoon region and most of the North Pacific) given the limited data storage capacity; hence, we are not able to evaluate the model performance globally. Also, we acknowledge that presenting global maps for evaluating the performance of UKESM1 may be useful (e.g. the global distributions of temperature, precipitation and aerosol optical depth); however, such an evaluation has been reported by Sellar et al. (2019) and reporting these analyses in this paper focusing on ARs and the East Asia region seems repetitive and out of scope.

In response to the comment, we have enlarged the geographical range of Figure S1 to better present the biases of the AR-related environmental fields over most of the North Pacific, Bay of Bengal and the East Indian Ocean. The study by Sellar et al. (2019) has been cited in lines 145-147 to help the reader locate the information on the performance of UKESM1.

Grey regions depict the Tibetan Plateau where the surface pressure is very likely below 850-hPa. A description of this has been added in the caption of Figure S1.

9) **Line 131:AR tracking was performed for each ensemble member, then the means were computed, correct?**

Authors:

That is correct as per the description in lines 132-136.

10) **Lines 150-170, AR detection: Thank you for this description. I did go to the supplemental material of Liang et al. 2022. I am assuming that the ARDT used in this work is the same ARIA-Asia described in this paper? If not, what are the differences and how do they compare to Figure S1 in that paper, i.e. other**

**ARTMIP ARDTs? I would highly recommend a version of that figure to be included in the supplemental material here given it is directly relevant. Because not all readers have the same level of accessibility to papers, it is best to have everything in one spot, especially when that information is so closely tied to the interpretation of results. Given this, I also recommend a more detailed description (including geometry, see Figure 7 comments) should be provided in the supplemental rather than relying on the Liang references?**

Authors:

As the reviewer mentioned, the used ARDT is named ARIA-Asia (lines 156-158) and the procedure of the used ARDT is the same as the schematic in Figure 1 of Liang et al. (2022).

As mentioned in lines 170-181, some adjustments (i.e. the thresholding method) of ARIA-Asia have been made but other configurations are the same as Liang et al. (2022). We agree with the reviewer that more details of the ARDT are needed; hence, more specific descriptions have been made in lines 165-177.

11) **Figure 1: Here and elsewhere, please darken the continental outlines, they are really hard to see?**

Authors:

Coastlines in all the spatial analysis figures have been darkened as suggested.

12) **Line 193: Seems like there are some words missing at the end of the sentence?**

Authors:

This sentence has been revised (line 202).

13) **Line 197: Is there more monsoonal precip here as well? The different SSP responses are interesting! For the aerosol feedbacks, is there a reference (such as the Jones et al 2021 you use later), or is this current work?**

Authors:

According to one of our ongoing studies, total precipitation and extreme precipitation events are projected to increase under G6sulfur over the regions affected by the East Asian summer monsoon flow. However, as AR features are isolated and discerned from the monsoon flow by the ARDT, changes in monsoon precipitation are not discussed here.

14) **Line 205: Shields et al., 2022 also shows that the dynamical AR climate/SAI change response is dictated by the jets?**

Authors:

We thank the reviewer for pointing out this. The study by Shields et al. (2022)

focusing on ARs affecting western North America has presented a similar weakening of the subtropical westerly jet (Figure 2a,b in Shields et al., 2022) but they did not discuss this in detail though close AR-jet relationships have been observed across western North America (e.g. Zhang and Villarini 2018).

15) **Line 219: with magnitudes (of weakening) greater than any other experiment?**

Authors:
  This has been removed to avoid any ambiguity.

16) **Figures 5,6,7: Unit should be labeled as (fraction of 6-hourly time steps) and not %.**

Authors:
  The unit of AR frequency in the shaded color scheme has been labelled as "Percent of Time Steps" (Figures 7, 8 and S1a-c).

17) **Line 256/257/324/345/356 and maybe elsewhere): Not sure "high latitude" is appropriate here, perhaps, upper-midlatitudes.**

Authors:
  These have been changed to "upper-midlatitudes" (lines 17, 19, 207, 281, 282, 287, 301, 332, 351, 354 and 364).

18) **Line 261: I am confused on the unit… % of time period, or fraction of time period? Percent is usually given in units 0-100%, whereas fraction is 0 to 1.**

Authors:
  As suggested, the unit of AR frequency has been changed to "Percent of Time Steps" (as reply to comment 16).

19) **Figure 7 and discussion:**
  **Are some of the geometrical qualities somewhat "baked into" the ARDT by definition (line 155)? The geometrical conditions are not specified in the AR detection section, but given this figure, I think they should be stated. Perhaps more detail in the ARDT in the supplemental material is needed, rather than relying on the given references.**
  **Although the discussion (I think) is using mostly the mean values and movement of the max and min whiskers, I am a little uncomfortable with these statements given that the \*range\* of possibilities within each category are quite similar for all panels except 7f. Some of the metrics have increased, or decreased, \*range envelopes\*, but that is about all you can say, especially given the small sample/ ensemble size. Please rewrite this section keeping in mind the uncertainties of each metric. The fact that many of these whiskers (for each**

**metric) completely overlap each other means that there is high uncertainty.**

**Line 287-288: AR frequency and size (as shown by the O'Brien paper) are very much tied to ARDT specifications. This, plus Fig 7, does not convince this statement can be made for anything over than moisture content for certain experiments.**

**Line 292-293: Same as above.**

**Line 351-352: An increase in wind speed along the AR core is not supported by Figure 7.**

Authors:

In response to these comments, we have removed this figure and the relevant discussion considering the limited information on the statistical significance presented by the error bars and the uncertainty of detecting these AR characteristics in the used ARDT. This also considers that these analyses are difficult to relate to the analyses of environmental mechanisms and AR-related precipitation. We agree that the description of the used ARDT should be more specific, especially for the used geometry criteria (see reply to comment 10).

20) **Figure 9 and 10: Because this area also experiences tropical cyclones, again, a brief sentence on how TCs are filtered out of the ARDT would be useful in the AR detection section (or supplemental). This may be for a different study, but it would be interesting to understand how much of the total precipitation is TC-related vs AR-related.**

Authors:

For the used ARDT, TC-like features with relatively short length and, sometimes, large width are removed by the thresholds of AR length (> 2000-km) and equivalent width (< 750-km). Relevant description on this has been added in lines 161-164.

In lines 388-391, we acknowledge that the contribution of ARs to precipitation is worth being compared with that from TCs to better understand the roles played by the different high-impact synoptic systems. However, the used UKESM1 simulations as well as most of the GeoMIP simulations are at a relatively coarse model resolution that is difficult to properly resolve the vortex structure of TCs. Diagnoses of TCs in such models usually rely on indices quantifying the genesis density of TCs by considering the large-scale thermodynamical fields (e.g. Webster et al., 2005; Jones et al., 2017; Emanuel 2021) and such a method is unable to discern TC-associated precipitation from the large-scale background. Thus, delivering the analysis of TC-associated precipitation in this study is technically difficult at present.

Thank you for your consideration of publication.

Sincerely yours,
Ju Liang (first author, corresponding author)
and
Jim Haywood

**Reference:**

Emanuel, K., 2005. Increasing destructiveness of tropical cyclones over the past 30 years. Nature 436, 686–8. https://doi.org/10.1038/nature03906

Jones, A., Haywood, J.M., Scaife, A.A., Boucher, O., Henry, M., Kravitz, B., Lurton, T., Nabat, P., Niemeier, U., Séférian, R., Tilmes, S., Visioni, D., 2022. The impact of stratospheric aerosol intervention on the North Atlantic and Quasi-Biennial Oscillations in the Geoengineering Model Intercomparison Project (GeoMIP) G6sulfur experiment. Atmos. Chem. Phys. 22, 2999–3016. https://doi.org/10.5194/acp-22-2999-2022

Jones, A.C., Haywood, J.M., Dunstone, N., Emanuel, K., Hawcroft, M.K., Hodges, K.I., Jones, A., 2017. Impacts of hemispheric solar geoengineering on tropical cyclone frequency. Nat. Commun. 2017 81 8, 1–10. https://doi.org/10.1038/s41467-017-01606-0

Kravitz, B., Robock, A., Tilmes, S., Boucher, O., English, J.M., Irvine, P.J., Jones, A., Lawrence, M.G., MacCracken, M., Muri, H., Moore, J.C., Niemeier, U., Phipps, S.J., Sillmann, J., Storelvmo, T., Wang, H., Watanabe, S., 2015. The Geoengineering Model Intercomparison Project Phase 6 (GeoMIP6): Simulation design and preliminary results. Geosci. Model Dev. 8, 3379–3392. https://doi.org/10.5194/GMD-8-3379-2015

Kravitz, B., MacMartin, D. G., Mills, M. J., Richter, J. H., Tilmes, S., Lamarque, J.-F., Tribbia, J. J., and Vitt, F.: First simulations of designing stratospheric sulfate aerosol geoengineering to meet multiple simultaneous climate objectives, J. Geophys. Res.-Atmos., 122, 12616–12634, https://doi.org/10.1002/2017JD026874, 2017.

Liang, J., Yong, Y., Hawcroft, M.K., 2022. Long-term trends in atmospheric rivers over East Asia. Clim. Dyn. 1–24. https://doi.org/10.1007/s00382-022-06339-5

MacMartin, D.G., Kravitz, B., Tilmes, S., Richter, J.H., Mills, M.J., Lamarque, J.F., Tribbia, J.J. and Vitt, F., 2017. The climate response to stratospheric aerosol geoengineering can be tailored using multiple injection locations. Journal of Geophysical Research: Atmospheres, 122(23), pp.12-574.

Webster, P.J., Holland, G.J., Curry, J.A., Chang, H.-R., 2005. Changes in tropical cyclone number, duration, and intensity in a warming environment. Science 309, 1844–6. https://doi.org/10.1126/science.1116448

Zhang, W., Villarini, G., 2018. Uncovering the role of the East Asian jet stream and heterogeneities in atmospheric rivers affecting the western United States. Proc. Natl. Acad. Sci. U. S. A. 115, 891–896. https://doi.org/10.1073/pnas.1717883115

---

## Author Response (AR2)

Dear Professor Geraint Vaughan,

Thank you for your acceptance of the manuscript for publication!

As suggested, a note "proxy for eddy kinetic energy and storm tracks" has been added to the caption of Figure 6 (lines 701-702) and the relevant description in the main text (line 243) following the comment from the reviewer.

Sincerely yours,

Ju Liang (first author, corresponding author)

Jim Haywood